# BKTyper: Free Online Tool for Polyoma BK Virus VP1 and NCCR Typing

**DOI:** 10.3390/v12080837

**Published:** 2020-07-31

**Authors:** Joan Martí-Carreras, Olga Mineeva-Sangwo, Dimitrios Topalis, Robert Snoeck, Graciela Andrei, Piet Maes

**Affiliations:** 1Zoonotic Infectious Diseases Unit, Laboratory of Clinical and Epidemiological Virology, Department of Microbiology, Immunology and Transplantation, Rega Institute, KU Leuven, Leuven BE3000, Belgium; joan.marti@kuleuven.be; 2Laboratory of Virology and Chemotherapy, Department of Microbiology, Immunology and Transplantation, Rega institute, KU Leuven, BE3000 Leuven, Belgium; olga.mineevasangwo@kuleuven.be (O.M.-S.); dimitrios.topalis@kuleuven.be (D.T.); robert.snoeck@kuleuven.be (R.S.); graciela.andrei@kuleuven.be (G.A.)

**Keywords:** Human BK polyomavirus (BKPyV), *VP1*, BKTGR, NCCR, genotyping

## Abstract

Human BK polyomavirus (BKPyV) prevalence has been increasing due to the introduction of more potent immunosuppressive agents in transplant recipients, and its clinical interest. BKPyV has been linked mostly to polyomavirus-associated hemorrhagic cystitis, in allogenic hematopoietic stem cell transplant, and polyomavirus-associated nephropathy in kidney transplant patients. BKPyV is a circular double-stranded DNA virus that encodes for seven proteins, of which Viral Protein 1 (VP1), the major structural protein, has been extensively used for genotyping. BKPyV also contains the noncoding control region (NCCR), configured by five repeat blocks (OPQRS) known to be highly repetitive and diverse, and linked to viral infectivity and replication. BKPyV genetic diversity has been mainly studied based on the NCCR and *VP1*, due to the high occurrence of BKPyV-associated diseases in transplant patients and their clinical implications. Here BKTyper is presented, a free online genotyper for BKPyV, based on a *VP1* genotyping and a novel algorithm for NCCR block identification. *VP1* genotyping is based on a modified implementation of the *BK typing and grouping regions* (BKTGR) algorithm, providing a maximum-likelihood phylogenetic tree using a custom internal BKPyV database. Novel NCCR block identification relies on a minimum of 12-bp motif recognition and a novel sorting algorithm. A graphical representation of the OPQRS block organization is provided.

## 1. Introduction

Human BK polyomavirus (BKPyV), or *Human polyomavirus 1*, was first described in 1971 by Gardner and collaborators [1]. Since its discovery, its prevalence has been increasing due to the introduction of more potent immunosuppressive agents, mostly in transplant-recipient patients. BKPyV has been mostly linked to two different transplantation diseases, polyomavirus-associated hemorrhagic cystitis (PyVHC) (5–15% of allogeneic hematopoietic stem cell transplant) [2] and polyomavirus-associated nephropathy (PyVAN) (1–10% of kidney transplant recipients) [3].

BKPyV is a circular double-stranded DNA virus (cdsDNA) with an average genome size of 5100 bp and an average GC content of 40%. Its genome is structured in two sections, the early and late coding regions. The early region encodes the regulatory proteins, i.e., large tumor antigen (LTag), small tumor antigen (sTag), and the truncated tumor antigen (truncTag), all derived by alternative splicing of a single primary transcript. The late region encodes the structural proteins VP1, VP2, and VP3, as well as a small regulatory protein known as Agno protein. Additionally, the BKPyV genome contains a noncoding control region (NCCR), also known as the regulatory region or the transcript control region, separating the early and late regions. The NCCR directs early and late transcription and replication of the genome as it contains the origin of replication (ori). NCCR is a bidirectional promoter-enhancer region that includes binding sites for several transcription factors [4]. The NCCR is known to be formed by repeat-rich regions, being highly variable in sequence and length [5,6,7]. Strains with NCCR rearrangements (i.e., deletions, insertions, or duplications of complete or partial blocks) are found in patients suffering from BKPyV disease. Previous studies have suggested the possible role of NCCR rearrangements in viral replication, as rearranged NCCR BKPyV emergence in plasma has been linked to increased replication capacity and disease in kidney transplant recipients [8].

Subtyping of BKPyV has been based on *VP1* genetic diversity. There are four major *VP1* subtypes: I, II, III, and IV. Typing of these regions has been first performed by restriction endonuclease of a 327 bp variable region of *VP1* [9]. Later, with the reduction of sequencing cost, routine typing of BKPyV has been conducted by Sanger sequencing and recently by Illumina sequencing [10]. Worldwide, the most abundant subtype is subtype I (80%), followed by subtype IV (15%) [2]. Contrary, the sister groups, i.e., subtypes II and III, are rarely detected. In its turn, subtype I can be subdivided in Ia, Ib-1, Ib-2, and Ic [11]; and subtype IV in IVa-1, IVa-2, IVb-1, IVb-2, IVc-1, and IVc-2 [12]. Epidemiology and geographical distribution of BKPyV have been previously studied [11], focusing on subgroups I and IV, which are known to have variant distributions between continents. BKPyV subtyping and subgrouping are conducted routinely in diagnostic assays and in epidemiological studies, albeit its prognostic value remains unclear. Recently, Morel et al. designed a strategy to subtype BKPyV, based on a 100 bp amplicon, called the *BK typing and grouping region* (BKTGR) [13]. In this same study, Morel et al. suggest a subtyping algorithm, validated through multiple sequence alignment and phylogeny.

The archetypical NCCR has been arbitrarily divided into diverse repeat-blocks as follows: O (142 bp), P (68 bp), Q (39 bp), R (63 bp) and S (63 bp). The O, P, Q, R, and S blocks, despite containing numerous transcriptional and binding sites, are not transcribed. BKPyV strains isolated and sequenced from urine in both immunocompromised and immunocompetent individuals mostly contain the archetypical NCCR architecture with minor variants, and it is thought to be the conformation found in transmissible virus [14,15].

It is worth noting that archetypical strains of BKPyV (being the WW strain the prototype strain), in contrast to rearranged types, replicate poorly in cell culture, indicating the role of NCCR rearrangements in viral growth in various cell types in vitro [16]. In addition, the rearranged structure of the NCCR are often found in kidney and other body compartments, linked to increased viremia, viruria [8,17], and frequently in association with diseases [18,19]. However, recent studies suggest that rearranged BKPyV is not required for developing PyVAN. Therefore, rearrangements appear not to correlate with viral reactivation disease and development, but with early and late gene expression and overall viral replication, coincidently with the high prevalence of transcriptional factor binding motifs [4,18,19]. It is hypothesized that rearranged viral strains indicate prolonged immunosuppression, favoring enhanced viral replication and therefore giving rise to NCCR rearrangements [4,18,19,20].

The incidence of BKPyV reactivation is mostly observed in renal transplant recipients, being a significant risk factor for developing PyVAN that is associated with a high chance for graft loss. Therefore, understanding BKPyV genetic factors that can be associated with increased pathogenicity is crucial. Due to the incidence of BKPyV-associated diseases and their clinical implications for human health, a reliable, automatic, and free BKPyV typing tool would be of great interest.

In this manuscript, BKTyper is presented as a reliable, automatic, free-typing tool for BKPyV virus genotyping based on *VP1* typing (implementing an adaptation of Morel et al. algorithm [13]) and a novel algorithm for reliable NCCR block identification. BKTyper can be found as an online free service (http://bktyper.zidu.be/) or on GitHub (https://github.com/joanmarticarreras/BKTyper) for local installation.

## 2. Materials and Methods

BKTyper has been implemented in Python3, using on the following libraries: Biopython (v1.73) [21,22], Numpy (v1.16.4) [23] and Pandas (v0.25.0rc0) [24]. BKTyper uses the following external software: Mafft (v7.429 2019/Jul/1) [25], BLAST (v 2.6.0+) [26] and IQTree [27]. The Biopython library contains the corresponding implementations for the Needleman–Wunch algorithm [28] and functions to launch external software and parse their results. Briefly, the input query is entered in (multi)fasta format, subsequently divided into single fasta entries and their orientation discerned, in comparison to RefSeq NCBI reference (polyoma BK Dunlop strain, NC_001538 or V01108.1), with blastn (-max_hsp 1, -evalue 0.000001). Sequences are reverse complemented if needed and their structure and coordinates rearranged to meet the reference. Genomes are posteriorly mapped with blastn (-word_size = 12, -evalue = 0.1 and -perc_identity = 75) against BKTyper custom NCCR BKPyV block archive (see Table 1) to discern coordinates, length and type of repeat. Additionally, BKTyper implements VP1 genotyping, applying an adaptation of the Morel et al. algorithm [13]. *VP1* is typed with Needlman–Wunch (gapopen = 10, gapextend = 0.5), aligning the query sequence against the Dunlop reference. Base content is checked at alignment positions: 1977, 1989, 1996, 2007, 2028, 2058, and 2076 (coordinates based on Dunlop strain), as devised by Morel et al. [13]. Mafft (v7.429 2019/Jul/1) [25] aligns, using default parameters, the VP1 gene from the query sequence to a custom VP1 database, producing a multiple-sequencing alignment (MSA). A maximum likelihood (ML) tree is computed at the BKTGR (BK typing and grouping region) using IQTree [27], with Mafft’s MSA. IQTree module ModelFinder [29] identifies the ML model that fits best to the information from the MSA, Kimura 2-Parameter (K2P+G4) model. Bootstrapping is conducted using IQTree module UFBoot2 [30] ultrafast bootstrapping. It is possible to type *VP1* and/or NCCR regions independently.

BKTyper can be found as an online free service (http://bktyper.zidu.be/) for automatic BKPyV typing and the source code available on GitHub (https://github.com/joanmarticarreras/BKTyper).

## 3. Results

BKTyper is the first free online tool that allows automatic and reproducible BKPyV typing. BKTyper conducts two independent, non-mutually exclusive, typings: (i) *VP1* genotyping, based on single nucleotide polymorphisms at the BKTGR region and (ii) NCCR structure identification, based on newly canonized OPQRS sequences, and a novel algorithm designed to discriminate incomplete blocks. Additionally, it provides the phylogenetic context for *VP1* BKTGR and graphical disposition for the NCCR structure.

### 3.1. VP1 Genotyping

The implementation of the *VP1* typing is an adaptation of the BKTGR algorithm proposed by Morel et al. in 2017 [13]. Such an algorithm needs to account for two premises: (i) variable length and coordinates from query sequences, and (ii) *VP1* gene or alignment can contain gaps.

#### 3.1.1. VP1 BKTGR (BK Typing and Grouping Region)

BKTGR typing relies on specific single nucleotide polymorphisms (SNPs) at reference positions 1976, 1988, 1995, 2006, 2018, 2057, and 2075 (*VP1* gene). Input sequences may have diverse lengths and may cover slightly different sections of the BKPyV genome. In order to generalize and automatize the typing process, BKTGR typing coordinates are first transferred to *VP1*-based coordinates (see Table 2). *VP1* alignments can have internal gaps, which will alter the correspondence between specific coordinates and nucleotides. Therefore, specific conserved DNA motifs from the reference are designed as that the first nucleotide corresponds to the BKTGR typing positions (see Table 2). Looking for specific conserved DNA motifs in the reference allows searching the BKTGR typing positions regardless of fixed numerical positions, which may change depending on sequence input. Alignments are posteriorly trimmed at both ends until the first alignment column without gaps, helping to standardize inputs composed of sequences of diverse lengths and reduce memory space. The complete implementation of the algorithm can be found in Appendix B, and a graphical representation in Figure 1.

#### 3.1.2. BKTGR Phylogeny

BKGTR subgroups and their genetic distances can be explored with ML phylogenetic tree. In order to build the tree, 60 sequences out of the 199 sequences present from Morel et al. [13], representing the 12 VP1 subgroups (KT354836.1, JX195578.1, JN794018.1, DQ989795.1, JN192436.1, AB301096.1, AY628225.1, JN794021.1, AB263932.1, KF468309.1, KT354835.1, JX195573.1, DQ989813.1, AB369092.1, JX195579.1, V01109.1, JF894228.1, DQ989812.1, KP984526.1, JN794032.1, DQ305492.1, V01108.1, AB211372.1, AB301098.1, AB211379.1, AB464954.1, AB369097.1, AB365165.1, Z19536.1, AB263920.1, JN793996.1, KF468295.1, JX195559.1, AB301101.1, M23122.1, JX195577.1, KF055892.1, JN192440.1, AB211386.1, AB269869.1, AB269859.1, AB365171.1, AB211389.1, AB269841.1, AB269826.1, AB211390.1, AB211391.1, AB217919.1, AB211387.1, AB269834.1, AB269846.1, AB365173.1, AB269863.1, AB269855.1, JN794007.1, JX195569.1, AB369093.1, JN794001.1, KT354834.1, KF468293.1, AB260034.1) are used. These sequences, together with the query sequence, are aligned with Mafft using default settings [25]. The alignment is posteriorly trimmed as explained earlier in the *VP1* BKTGR section. This alignment, consisting of approximately 107 nucleotides, is modeled using ModelFinder [29], identifying Kimura 2-Parameter (K2P+G4) as the best performing ML model. The phylogenetic tree is built with IQTree [27] using 10,000 ultrafast bootstraps from IQTree module UFBoot2 [30] as faster and more accurate surrogate for classical bootstrapping. Tree is rooted by mid-point distance of the tree.

#### 3.1.3. VP1 Genotyping Validation

BKTyper can correctly classify the 199 BK polyomavirus genomes used in Morel et al. [13], including recombinants VP1 JX195567 and JX195570, into Ia, Ib-1, Ib-2, Ic, II, III, IVa-1, IVa-2, IVb-1,2, IVc-1, and IVc-2 genotypes. A subset of those sequences is used as an internal database for the BKTyper ML tree. As can be observed in Figure 1B, the phylogenetic clustering of the different subtypes is conserved, validating the implementation of the algorithm.

### 3.2. NCCR Typing

Until now, NCCR typing has been mainly conducted by manual curation. This approach tends to be very tedious and prone to error. However, in order to optimize this process, two main aspects must be considered for its automatization: (i) canonization of the archetypical OPQRS blocks, and (ii) length and diversity of the blocks between isolates.

#### 3.2.1. Defining a Canon for the Archetypical OPQRS Blocks

Over the years, several original works have tackled the content and diversity of the OPQRS block [5,7,31,32]. Initially, Markowitz et al. defined nucleotide block length as per P(68), Q(39), and R(63) [31]. Later, Moens et al. described the O block, which contains the origin of replication, the TATA box for early genes, and several putative promoters. The same article also described S block, elapsing until the start of the *agno* gene. The block length defined by Moens et al. using 18 sequence consensus is O(142), P(68), Q(39), R(63), and S(63) [33], albeit it is also stated as a P(68) for the archetypical polyomavirus BK MM strain [32]. Later, Burger-Calderón et al. stated the block length as O(142), P(70), Q(39), R(40), and S(64) [7]. Here we propose a variant for the arbitrary classification of the NCCR blocks, considering preceding efforts, in order to improve, automatize and harmonize its typing. Several inconsistencies between Moens et al. and Burger-Calderón et al. block definition and its subsequent correction in BKTyper archive are later highlighted (Figure 2): (i) In Burger-Calderón et al., the first T of the O block is omitted (the first nucleotide of the origin of replication). (ii) In Burger-Calderón et al., the first G of the Q block is missing. (iii) In Moens et al., there is a discrepancy on P length between Figure 1 [P(68)] and Figure 2 [P(107)]. In Figure 2 of the manuscript, the P block seems to carry an identical copy of the Q block at the end. Finally, (iv), in Burger-Calderon et al., the R block is an exact copy of the Q block with an extra A at the end.

#### 3.2.2. NCCR Typing Based on Local Alignment

Once the NCCR blocks sequence content is standardized, there are additional premises to consider for its implementation in an automatic tool: (i) blocks will generally follow an OPQRS structure. (ii) Blocks can be repeated in tandem (i.e., OPPQRS). (iii) Blocks can be non-tandem repeated (i.e., OPQPRS). (iv) Block duplication may be incomplete [i.e., P(20) vs. P(70)]. (v) NCCR blocks are not coding and are only functional in short transcriptional binding sites, therefore, high diversity between input sequences is expected. (vi) Terminal regions of a block with low similarity with the references may be excluded from the block. (vii) NCCR blocks may be missing. Subsequently, alignments must allow for zero or more hits of each NCCR fragment per query, accounting for diverse order and minimum alignment size and nucleotide identity. Minimum alignment size has been delimited to 12 exact nucleotides, meaning that at least 12 exact nucleotides are needed between the query sequence and the NCCR archive block sequences in order to progress the classification. This parameter has been manually fine-tuned in order to ensure a low false-positive rate and a high sensitivity for fragmented blocks. Minimum identity between a putative block in the query sequence and a given block from the NCCR archive sequences has been set to 75%, 5% more stringent than previous estimates [7], as an arbitrary but reasonable similarity threshold for noncoding region. Based on these considerations, NCCR typing is conducted by locally aligning the query nucleotide sequence to a custom NCCR archive (see Table 1). Blast results are filtered by a minimum e-value of 0.05. Results are ordered by start sequence position and for each alignment identifiable as a putative NCCR block, start and end alignment coordinates are stored. For each non-overlapping alignment, the longest alignment is classified as a NCCR block. The complete implementation can be found in Appendix C and a graphical representation on Figure 3.

BKTyper includes a visual inspection of NCCR block structure as well as a short list of conserved transcriptional binding sites that can be found as exact matches from Moens et al. (see Table 3) [33].

#### 3.2.3. NCCR Typing Validation

Validation of NCCR BKTyper has been carried out with the 3 most used reference genomes (Gardner ATCC VR-837, MM, and Dunlop strains) and the 10 available sequences reported by Carr et al. [5]. BKTyper identifies the NCCR structure for the Gardner strain (LC029411.1) as OPQPQRS, instead of OPQPQS (as seen in Figure 4). Interestingly, MM strain typing (V01109.1) is OPQPQPQS, instead of OPQPQQS, as referenced elsewhere [7] (see Figure 4) and Dunlop (V01108.1) as OPPPS, instead of OPPS, as previously referenced [7] (see Figure 4). MM strain blocks (see Figure 4) reveals a partial P block of only 37 nucleotides between the two last Q blocks, corresponding to the end of the P block. Likewise, the Dunlop strain (Figure 4), contains an extra P block of 34 nucleotides between 2 complete P blocks. Visual inspection of the alignments verifies the re-typing of the classical strains (see Figure 4). Additionally, seminal work by Markowitz et al. [31] independently corroborates BKTyper NCCR classification. An attentive look at Figure 1 from Markowitz et al. (page 3390), displays the PQR block structure of the WW, Dunlop, and MM strains. Furthermore, in Markowitz et al. it is stated that “BK(Dunlop) could have arisen from BK(WW) by deletion of the Q and R blocks and 4 bp of flanking DNA, triplication of the P block and subsequent deletion of 18 bp within the second of the three repeated P blocks.” [31]. Other works, such as in Bethge et al., also refer to a P block triplication in Dunlop strain [4]. Similarly, for MM strain, in Markowitz et al., there is evidence for a triple repeat of the PQ junction, as in “The first step could have involved the deletion of the rightmost 55 bp of the r block, together with 12 bp of the adjacent unique sequence. A unit encompassing 53 bp of the P block and 34 bp of the Q block may have then have been duplicated, with another 4 bp of unique sequence inserted between the repeats. Another duplication encompassing 36 bp of the P block and 25 bp of the Q block may then have occurred.”

Different block identification has been reported between authors, articles, or reports. This lack of concordance may have occurred (i) due to alternative viral passage history (i.e., OPQRS configuration between Burger-Calderón et al. [7,34] - BKPyV ATCC 45026- and Yang et al. [35] -original MM strain-), or (ii) differences in stringency on pattern recognition (i.e., number of bp needed to identify a block, as with BKPyV Gardner strain).

In Carr et al., 10 sequences were presented, of which WW (OPQRS), SJH16A (OPQPQRS), SJH16B (OPQRS), SHJ18A (OPQPQRS), SHJ18C (OPQP), SHJ85A (OPQPQRS), SHJ85B (OPQRS), MAN10B (OPQRS) and MAN11B (OPQRS) have their NCCR architecture correctly predicted by BKTyper, as can be observed in Figure 5 [5].

BKTyper has been used to genotype all available full-length genomes for BKPyV listed as genome neighbors from NIH NCBI resource. Out of 318 genomes (accession numbers in Supplementary Information Appendix A, March 2020), 20 genomes were not suited for BKTyper complete genotyping, as lack the origin of replication and part of the NCCR. Those were later resubmitted for *VP1* typing alone. Complete results are available as Supplementary Information Appendix A. Out of 318 sequences, 95 represent subgroup Ic (29.87%), 64 Ib-2 (20.13%), 45 Ib-1 (14.15%), 29 IVc-2 (9.12%), 23 Ia (7.23%), 20 as IVb-1 or IVb-2 (6.29%), 17 IVc-1 (5.35%), 8 III (2.52%), 7 as IVa-2 (2.20%), 5 IVa-1 (1.57%), 5 as II subgroups (1.57%). Regarding diversity of NCCR organizations, 267 represent the canonical OPQRS (83.96%), 20 (SI 2) do not present origin of replication and/or compete NCCR (6.29%), 10 OPPQRS (3.14%), 4 OPQROPQRS (1.26%), 4 OPQPQS (1.26%), 4 OPOPQRS (1.26%), 2 OPPPS (0.63%), 1 OPS (0.314%), 1 OPQSPQS (0.314%), 1 OPQRSRS (0.314%), 1 OPQQRS (0.314%), 1 OPQPQPQS (0.314%), 1 OPQOPQRS (0.314%) and 1 OPPQR (0.314%). Combined genotyping, as seen in Table 4, shows the abundance distribution of NCCR structure and *VP1* groups, mostly represented by archetypical NCCR structure and *VP1* groups I and IV.

BKTyper provides a free, automated, and reproducible alternative to manual genotyping of *VP1* and NCCR for BKPyV. It has shown to be sensible enough to detect inconsistencies in previous literature and perfect to summarize genotyping information in the range of hundreds of sequences in a few minutes. BKTyper will allow researchers and clinicians to obtain BKPyV typing in a fast and reliable manner, bolstering its research and clinical forecasting during routine screenings.

## Figures and Tables

**Figure 1 viruses-12-00837-f001:**
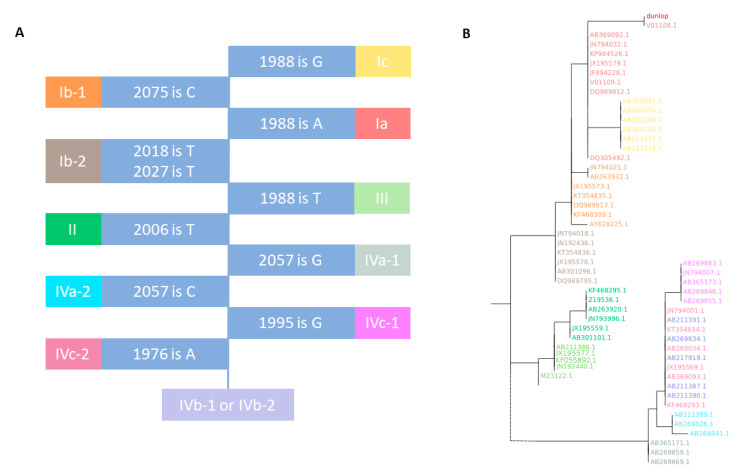
Graphical representation of BKTyper *VP1* genotyping and phylogeny block identification. Subpanels (**A**) and (**B**) represent the different steps of the algorithm. (**A**), decision tree to genotype *VP1* from Morel et al. [13]. Blue boxes represent coordinates for specific positions from the polyomavirus BK Dunlop strain (V01108.1) reference and the nucleotide that needs to be present to stop the decision tree. If true, the subgroup corresponds to the contiguous colored box. If another nucleotide is found at the given position, the following colored box must be checked. (**B**), ML phylogenetic tree that corresponds to the typing of a Dunlop strain *VP1* with the internal database of BKTyper. Colors represent the different subtypes (shared in panel **A**). Query sequence “dunlop” is highlighted in red.

**Figure 2 viruses-12-00837-f002:**
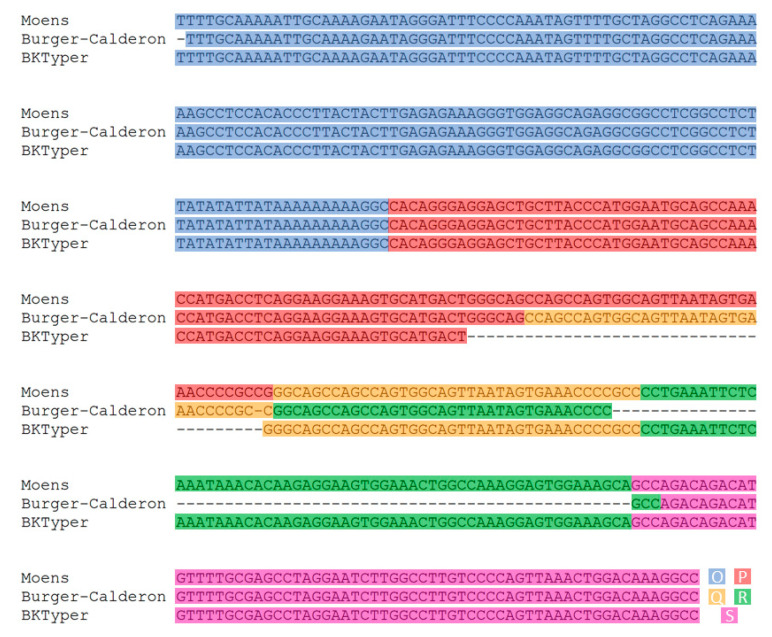
Visual inspection of the NCCR sequence alignment between Moens et al., Burger-Calderón et al., and BKTyper (see Table 1) [7,33]. Editing errors from both Moens et al. and Burger-Calderón et al. are displayed in the alignment. This alignment was constructed by concatenating the different OPQRS NCCR blocks and posteriorly aligned with Mafft [25]. Diverse color underlying has been used to highlight the different blocks (blue for O, red for P, yellow for Q, green for R, and purple for S).

**Figure 3 viruses-12-00837-f003:**
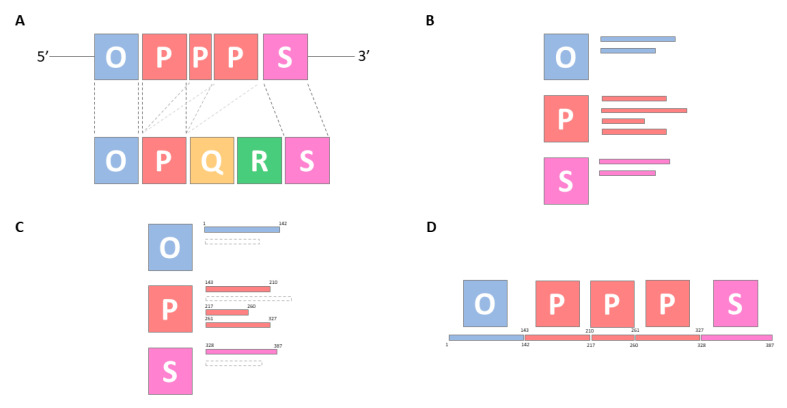
Graphical representation of BKTyper NCCR block identification using BKPyV Dunlop strain (V01108.1). Subpanels (**A**) to (**D**) represent the different steps of the algorithm. (**A**) Alignment of the query sequence (upper blocks) to the BKTyper NCCR archive sequences (bottom blocks). (**B**) Collection of alignment hits (small rectangles) by block. Alignment hits can represent (i) off-target hits or (ii) sub-alignments (as displayed by arbitrary rectangles in subpanel (**B**). (**C**) Alignment hit filtering by similarity and alignment coordinates. (**D**) Reconstruction of the original sequence and typing reporting. NCCR regions are codified as blue for O, red for P, yellow for Q, green for R, and purple for S blocks.

**Figure 4 viruses-12-00837-f004:**
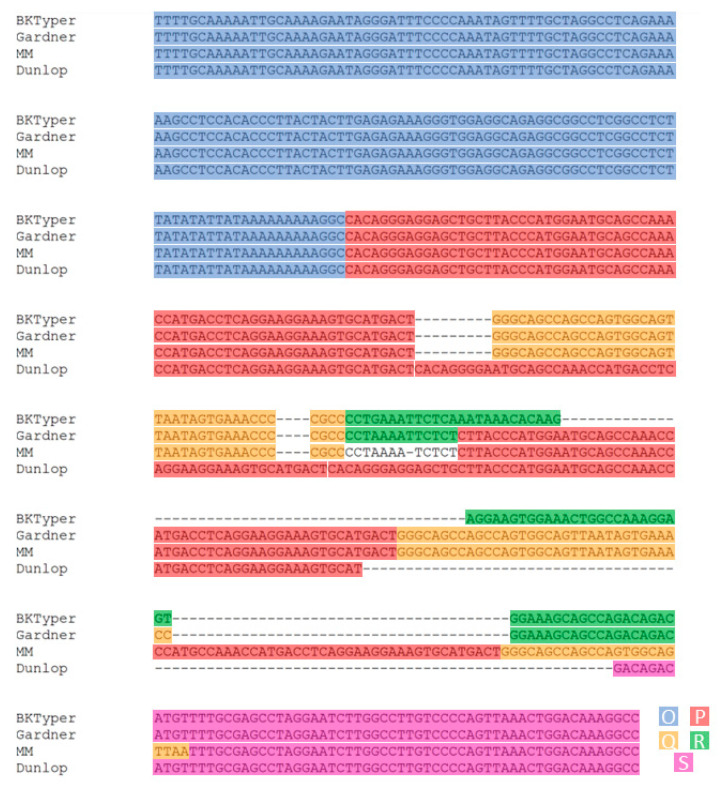
Visual confirmation of the NCCR sequence alignment of Gardner (LC029411.1), MM (V01109.1), and Dunlop (V01108.1) strains. This alignment was constructed by concatenating the different OPQRS NCCR blocks and posteriorly aligned with Mafft [25]. Diverse color underlying has been used to highlight the different blocks (as blue for O, red for P, yellow for Q, green for R, and purple for S).

**Figure 5 viruses-12-00837-f005:**
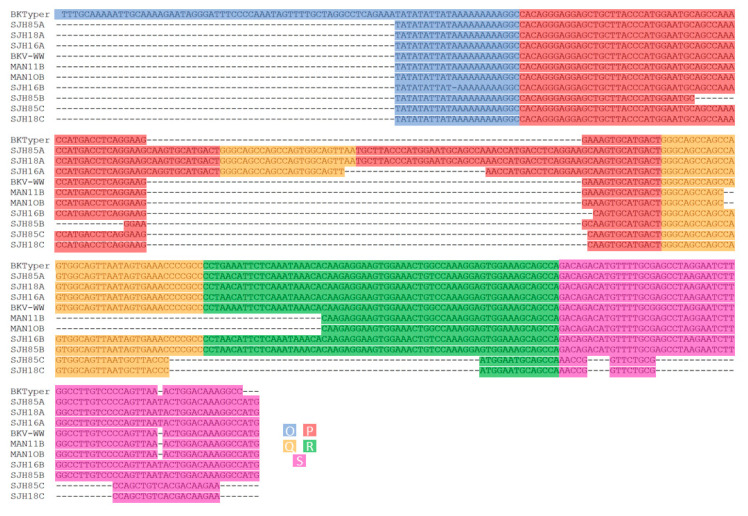
Visual confirmation of the NCCR sequence alignment Carr et al. [5]. This alignment was constructed by concatenating the different OPQRS NCCR blocks and posteriorly aligned with Mafft [25]. Diverse color underlying has been used to highlight the different blocks (blue for O, red for P, yellow for Q, green for R, and purple for S).

**Table 1 viruses-12-00837-t001:** The complete sequence of the OPQRS blocks used as archetypes for the BKTyper NCCR archive.

NCCR Block	Sequence
O	TTTTGCAAAAATTGCAAAAGAATAGGGATTTCCCCAAATAGTTTTGCTAGGCCTCAGAAAAAGCCTCCACACCCTTACTACTTGAGAGAAAGGGTGGAGGCAGAGGCGGCCTCGGCCTCTTATATATTATAAAAAAAAAGGC
P	CACAGGGAGGAGCTGCTTACCCATGGAATGCAGCCAAACCATGACCTCAGGAAGGAAAGTGCATGACT
Q	GGGCAGCCAGCCAGTGGCAGTTAATAGTGAAACCCCGCC
R	CCTGAAATTCTCAAATAAACACAAGAGGAAGTGGAAACTGGCCAAAGGAGTGGAAAGCAGCCA
S	GACAGACATGTTTTGCGAGCCTAGGAATCTTGGCCTTGTCCCCAGTTAAACTGGACAAAGGCC

**Table 2 viruses-12-00837-t002:** BKTGR coordinates correspondence and its specific motifs. Genome coordinates are based on polyoma BK Dunlop strain (V01108.1) reference. *VP1* coordinates correspond to nucleotide coordinates of the same reference using the first base of the gene as a starting position. The conserved motif corresponds to the motif used to transport coordinates from the reference to the query sequence (first nucleotide of the motif).

Genome Coordinates	VP1 Coordinates	Conserved Motif
1989	426	AAAACCTAT
2076	513	AAGTAC
2019	456	CTTTGCTG
2028	465	AGGTGGAGAA
2007	444	TAATTTCCACTTCTTTG
2058	495	GCTAATGAATTACAG
1996	433	ATTCAAGGCAGTAATTT
1977	414	GCATGGTGGAGGAAA

**Table 3 viruses-12-00837-t003:** Shortlist of transcriptional binding from Moens et al. [33] that are included in BKTyper NCCR graphical representation. The first column displays the name of the transcriptional sites, followed by its sequences, as shown in Moens et al. [33].

Transcriptional Binding Sites	Motif
Promoter IL-6 gene	TTCC
T-Antigen	GCCTC or GCCCC
NF-1	TCCA or TGGCCTTGTCCCCAG
Polyomavirus enhancer B	AGAGG
SP-1	AGGCGG
Unknown JC polyomavirus binding factor (JVC)	GGGAGGAG
Cytomegalovirus immediate early promoter (CMV IE-1)	GGAAAG
NFkB	GTGAAACCCC
SV40 enhancer-core	TGGAAAG
CRE	TGACCTCA
GRE	TGTCCC
Murine Thy-1	AGGC
TATA box	TATAA
Transcription factor Late SV40 (LSF)	CCCGCC

**Table 4 viruses-12-00837-t004:** The abundance of BKPyV genome neighbors based on combined NCCR and *VP1* genotyping. Percentage, absolute number of sequences (N), NCCR structure, and *VP1* group are provided as columns. NA denoted not available.

Percentage	N	NCCR	VP1
24.53	78	OPQRS	Ic
16.04	51	OPQRS	Ib-2
12.26	39	OPQRS	Ib-1
9.12	29	OPQRS	IVc-2
5.35	17	OPQRS	IVc-1
5.35	17	OPQRS	IVb-1,2
4.09	13	NA	Ib-2
3.14	10	OPQRS	Ia
2.52	8	OPQRS	III
2.52	8	OPPQRS	Ic
2.20	7	OPQRS	IVa-2
1.57	5	OPQRS	IVa-1
1.57	5	OPQRS	II
1.26	4	OPQROPQRS	Ic
1.26	4	OPQPQS	Ia
1.26	4	OPOPQRS	Ic
1.26	4	NA	Ia
0.94	3	NA	Ib-1
0.63	2	OPPQRS	Ib-1
0.63	2	OPPPS	Ia
0.31	1	OPS	Ia
0.31	1	OPQSPQS	Ia
0.31	1	OPQRSRS	IVb-1.2
0.31	1	OPQQRS	Ib-1
0.31	1	OPQPQPQS	Ia
0.31	1	OPQOPQRS	Ic
0.31	1	OPPQR	IVb-1.2

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
