# Peer review of "BKTyper: Free Online Tool for Polyoma BK Virus VP1 and NCCR Typing"

_viruses, 2020, doi:10.3390/v12080837_

Round 1
Reviewer 1 Report
Marti-Carreras and co-workers here present an online tool here to genotype the BKPyV from based on the NCCR and VP1. The algorithm used appears to be a published BKTGR workflow, with the addition of an internal database of BkPyV sequences.
I believe the online GUI (or at least images of it, since the website http://bktyper.zidu.be/ is not up at the time of this review) will in theory allow for a useful and easy interface for rapid genotyping of BKPyVs.
I have a few minor comments that I hope the authors can address:
1. Please spell out all abbreviations (eg PVAN in line 76)
2. The authors should be more detailed in how their current implementation differs from the one published from the Morel paper (BKTGR). Eg input sequences, interfaces, outputs etc. The authors should take note in laying out the NOVELTY of this implementation, other than just a GUI interface for BKTGR.
3. Code for the implementation should be deposited onto Github and made available to the public.
4. It would be helpful to have a graphical representation of the algorithm that BKTyper goes through, in a flow chart format, all the way from the initial FASTA file input to the outputs. Essentially, this lays out the Appendixes A and B in a rapidly reader digestible way, and would be appropriate as the initial figure.
Thank you for your work and time, and the opportunity to review your work.
Sizun Jiang
Author Response
Marti-Carreras and co-workers here present an online tool here to genotype the BKPyV from based on the NCCR and VP1. The algorithm used appears to be a published BKTGR workflow, with the addition of an internal database of BkPyV sequences.
I believe the online GUI (or at least images of it, since the website http://bktyper.zidu.be/ is not up at the time of this review) will in theory allow for a useful and easy interface for rapid genotyping of BKPyVs.
There have been recent issues with KU Leuven (our university) server administration due to high user overload (exams placed on-line and an increase in teleworking). Unfortunately, our server was affected. At the time of this re-submission, the site http://bktyper.zidu.be/ should be online, sorry for the inconvenience and thank you for your patience.
I have a few minor comments that I hope the authors can address:
- Please spell out all abbreviations (eg PVAN in line 76)
Instead of PVAN should read PyVAN (now line 77), which is fully spelled out at line 34-35.
- The authors should be more detailed in how their current implementation differs from the one published from the Morel paper (BKTGR). Eg input sequences, interfaces, outputs etc. The authors should take note in laying out the NOVELTY of this implementation, other than just a GUI interface for BKTGR.
Thank you for your comment. We appreciate that it is important to highlight the novelty aspect of our research. BKTyper is novel as it is the first and only free GUI for polyoma BK typing. It provides the first algorithm and automatic implementation for NCCR identification and classification. Additionally, it provides the first automatic implementation for BKTGR algorithm. Abstract and introduction have been improved, highlighting the novelty of BKTyper.
- Code for the implementation should be deposited onto Github and made available to the public.
We agree with the reviewer on the importance of open source research, both during reviewing processes and publication. BKTyper - v.0.1 is available at https://github.com/joanmarticarreras/BKTyper
- It would be helpful to have a graphical representation of the algorithm that BKTyper goes through, in a flow chart format, all the way from the initial FASTA file input to the outputs. Essentially, this lays out the Appendixes A and B in a rapidly reader digestible way, and would be appropriate as the initial figure.
Thank you very much for your comment. In fact, we had an initial figure laying out the structure of the program. It happened to be too crowded and difficult to read. Therefore, it was decided to split it into Figure 1 and Figure 3 (BKTGR and NCCR analysis respectively).
5. Thank you for your work and time, and the opportunity to review your work.
Thank you for your comments and your contribution to this manuscript. We appreciate your constructive criticism and the idea to add our program to GitHub. It might increase its use by being integral part in BKPyV NGS pipelines.
Reviewer 2 Report
In the article “BKTyper: Free online tool for Polyoma BK virus VP1 and NCCR typing”, Martí-Carreras et al. present a free online tool that uses the algorithm recently published by Morel et al. (https://doi.org/10.1128/JCM.01180-16) to determine the genotype of BKPyV strains. Most interestingly, this tool also enables to determine the structure of the Non-Coding Control Region (NCCR). This region of the BKPyV genome can have an archetypal structure with five domains (O, P, Q, R and S) but can also be rearranged in some strains after mutation(s), deletion(s) or duplication(s). Actually, the determination of the NCCR structure is mostly performed by aligning sequences manually but this frequently leads to NCCR annotation differences from one lab to another.
This tool has been presented at “Viruses 2020 - Novel Concepts in Virology” in Barcelona (5 - 7 February 2020) with an abstract in “Proceedings” (https://doi.org/10.3390/proceedings2020050025) and I think that it could be very useful for the BKPyV community. Unfortunately, it is not yet accessible online and thus it could not be evaluated. I also think that several data should be presented in the “Results” section instead of the “Discussion”. Furthermore, I found it regrettable that the manuscript has not been proofread carefully. I thus propose to reconsider the manuscript after major revision.
Specific comments :
L15-16 : It is mentioned in the abstract that “BKPyV is a circular double-stranded DNA virus that encodes for 5 proteins”. However, as mentioned L38-40, there are the LTag, sTag, truncTag, VP1, VP2, VP3 and Agno proteins.
L44-46 : “NCCR is as bidirectional promoter-enhancer region that includes binding sites for several transcription factors (as seen in Figure 1)”. However, this is not shown in the Figure 1.
Table 1 : It is not clear why the authors chose an R domain of 60 bp whereas the size of 63 bp is largely accepted in the BKPyV community.
Figure 1 panel B : The operation performed at this step is not clear. Why 2 O blocks, 4 P blocks and 2 S blocks are recognized ?
L126 and L151 : There are two titles “3.1.1”.
L180 : “The block length defined by Moens et al. using 18 sequence consensus is O(143), P(109), Q(39), R(61) and S(64) [29], albeit it is also stated a P(68) for the archetypical polyomavirus BK MM strain [29]”. Contrary to what is stated in this reference, Moens et al. defined the block length as O(142), P(68), Q(39), R(63) and S(63) (cf figure 1 of this reference).
L181-182 : What is the reference for Burger-Calderon et al. ?
L218 : “The complete implementation can be found in Appendix B and a graphical representation on Figure 3”. Contrary to what is stated in this sentence, the figure 3 is not related to NCCR identification.
Table 3 : Please proofread carefully this table. For instance, I think there is a mistake for “SV40 T-antigen / TGGAAAG”.
L246 : “MM strain blocks (Fig. 3) reveals a partial P block of only 37 nucleotides between the two last Q blocks, corresponding to the end of the P block”. However, this is not shown in the Fig.3.
L256-259 : Please clarify
The discussion contains several data with one table and two figures. It should have been better to include these results in the “Results” section and to discuss these results in the “Discussion”.
The colours of the figures are very pale.
L52 : “major” instead of “mayor” ?
L54 : “BKPyV” instead of “BKV” ?
L75 : The reference 16 does not seem to be adapted.
L88 : “based on VP1 typing” instead of “for based on VP1 typing” ?
L253 : “[…]” The full sentence should be cited to avoid the reader having to go back and read the original reference.
L253 : “Bethge” instead of “Bethege”.
Author Response
In the article “BKTyper: Free online tool for Polyoma BK virus VP1 and NCCR typing”, Martí-Carreras et al. present a free online tool that uses the algorithm recently published by Morel et al. (https://doi.org/10.1128/JCM.01180-16) to determine the genotype of BKPyV strains. Most interestingly, this tool also enables to determine the structure of the Non-Coding Control Region (NCCR). This region of the BKPyV genome can have an archetypal structure with five domains (O, P, Q, R and S) but can also be rearranged in some strains after mutation(s), deletion(s) or duplication(s). Actually, the determination of the NCCR structure is mostly performed by aligning sequences manually but this frequently leads to NCCR annotation differences from one lab to another.
This tool has been presented at “Viruses 2020 - Novel Concepts in Virology” in Barcelona (5 - 7 February 2020) with an abstract in “Proceedings” (https://doi.org/10.3390/proceedings2020050025) and I think that it could be very useful for the BKPyV community. Unfortunately, it is not yet accessible online and thus it could not be evaluated. I also think that several data should be presented in the “Results” section instead of the “Discussion”. Furthermore, I found it regrettable that the manuscript has not been proofread carefully. I thus propose to reconsider the manuscript after major revision.
There have been recent issues with KU Leuven (our university) server administration due to high user overload (exams placed on-line and an increase in teleworking). Unfortunately, our server was affected. At the time of this re-submission, the site http://bktyper.zidu.be/ should be on-line. Thank you for your patience.
As a methodological paper or a presentation of an on-line resource/program, it can be difficult to layout a clear cut between “Results” and “Discussion”. We tried to present how the 2 algorithm implementations work and their outcomes in “Results”. Meanwhile in “Discussion” we tried to put the results into context, by comparing the genotype prediction with prior work for both BKTGR typing and NCCR. In this section we decided to add an unexpected result, the misclassification of classical BKPyV strains MM, Gardner and Dunlop, which was found by literature discussion. We agree with the reviewer that this later section may be suitable for “Results”. As Viruses – MDPI allows combination of “Results” and “Discussion” sections (https://www.mdpi.com/journal/viruses/instructions), we decided to re-order “Discussion”. “Results” sections 4.1 and 4.2 have been merged with “Discussion” as 3.1.3 and 3.2.3, respectively.
Specific comments:
L15-16 : It is mentioned in the abstract that “BKPyV is a circular double-stranded DNA virus that encodes for 5 proteins”. However, as mentioned L38-40, there are the LTag, sTag, truncTag, VP1, VP2, VP3 and Agno proteins.
The typo has been corrected in the abstract as “7” instead of “5” (line 15).
L44-46 : “NCCR is as bidirectional promoter-enhancer region that includes binding sites for several transcription factors (as seen in Figure 1)”. However, this is not shown in the Figure 1.
Corrected to “NCCR is as bidirectional promoter-enhancer region that includes binding sites for several transcription factors [4].”. Line 44-45.
- Bethge, T.; Hachemi, H.A.; Manzetti, J.; Gosert, R.; Schaffner, W.; Hirsch, H.H. Sp1 Sites in the Noncoding Control Region of BK Polyomavirus Are Key Regulators of Bidirectional Viral Early and Late.
Table 1 : It is not clear why the authors chose an R domain of 60 bp whereas the size of 63 bp is largely accepted in the BKPyV community.
R block should consist of 63 bp, as in Moens et al. Table 1 was depicting a first version of the NCCR sequences for BKTyper. These first sequences were shorter as to improve sequence alignments and block identification. Table 1 depicts the exact sequence that it is used for BKTyper (the same as in Moens et al.).
Figure 1 panel B : The operation performed at this step is not clear. Why 2 O blocks, 4 P blocks and 2 S blocks are recognized ?
Figure 1 is now Figure 3. There, a schematic of the algorithm for NCCR block recognition and classification is depicted. Small unlabeled rectangle account for alignment hits. After aligning with BLAST against multiple sequences, it is possible to obtain several hits either for different sequences or different sub-alignments for the same hit. The 2O, 4P and 2S blocks are arbitrary numbers of sub-alignments. Further details are given in the figure legend in order to avoid further confusion.
L126 and L151 : There are two titles “3.1.1”.
Thank you for pointing this out, line 161 has now been changed to “3.1.2 BKTGR phylogeny“.
L180 : “The block length defined by Moens et al. using 18 sequence consensus is O(143), P(109), Q(39), R(61) and S(64) [29], albeit it is also stated a P(68) for the archetypical polyomavirus BK MM strain [29]”. Contrary to what is stated in this reference, Moens et al. defined the block length as O(142), P(68), Q(39), R(63) and S(63) (cf figure 1 of this reference).
Now line 195. Block length has been corrected as per the contents of Figure 3 of Moens et al.
L181-182 : What is the reference for Burger-Calderon et al. ?
Reference to Burger-Calderón et al. 2016 has been added to line 198. The reference is used to compare the block size between previous literature (Moens et al.) and BKTyper.
L218 : “The complete implementation can be found in Appendix B and a graphical representation on Figure 3”. Contrary to what is stated in this sentence, the figure 3 is not related to NCCR identification.
Amended with the rearrangement of text sections.
Table 3 : Please proofread carefully this table. For instance, I think there is a mistake for “SV40 T-antigen / TGGAAAG”.
Table 3 has been proofread and updated to SV 40 enhancer-core.
L246 : “MM strain blocks (Fig. 3) reveals a partial P block of only 37 nucleotides between the two last Q blocks, corresponding to the end of the P block”. However, this is not shown in the Fig.3.
On line 254 now reads “MM strain block (see Figure 4)”.
L256-259 : Please clarify
Lines 270-274. We agree with the reviewer that this fragment may require further elaboration. The text has been expanded to show full in-text citation from Markowitz et al. on MM strain and the final remarks have been re-written for clarity.
The discussion contains several data with one table and two figures. It should have been better to include these results in the “Results” section and to discuss these results in the “Discussion”.
We agree with the reviewer that “Discussion” contains sections more akin to “Results”. The authors decided to combine both “Results” and “Discussion”. More information on our reasoning is provided in the initial reply to the reviewer.
The colours of the figures are very pale.
Alignments have been repeated and the color palette has been changed to more vibrant colors. Color palette is still conserved between Figures 2, 3, 4 and 5.
L52 : “major” instead of “mayor” ?
Corrected at line 52.
L54 : “BKPyV” instead of “BKV” ?
Corrected at line 54.
L75 : The reference 16 does not seem to be adapted.
Now as reference 17 has been updated to Wang, R.Y.L.; Li, Y.-J.; Lee, W.-C.; Wu, H.-H.; Lin, C.-Y.; Lee, C.-C.; Chen, Y.-C.; Hung, C.-C.; Yang, C.-W.; Tian, Y.-C. The association between polyomavirus BK strains and BKV viruria in liver transplant recipients. Sci. Rep. 2016, 6, 28491.
L88 : “based on VP1 typing” instead of “for based on VP1 typing” ?
Amended at line 89.
L253 : “[…]” The full sentence should be cited to avoid the reader having to go back and read the original reference.
Expanded at lines 261-269.
L253 : “Bethge” instead of “Bethege”.
Amended at line 263.
Thank you for your thorough review and insightful comments on this manuscript. This review has substantially improved our work.
Round 2
Reviewer 2 Report
Dear authors, congratulations for the development of BKTyper. As previously mentioned, I think that this tool will be very useful for the BKPyV community. You will find below some comments on the V2 of your manuscript as well as your tool :
I still have some concerns concerning the section 3.2.1 and the Figure 2. As mentioned previously, Moens et al. defined the block length as O(142), P(68), Q(39), R(63) and S(63) (cf figure 1 of this reference), which is perfectly similar to the sequence used for BKTyper. Thus, there should be no misalignement between Moens and BKTyper in the Figure 2. Concerning the sequence mentioned by Burger-Calderon et al., I found that the block length was O(141), P(69), Q(38), R(39) and S(63). Thus:
- “(i) In Burger-Calderón et al. the first T of the O block is omitted (the first nucleotide of the origin of replication).” → OK
- “(ii) In Burger-Calderón et al., the first G of the Q block is missing.” → indeed it is placed at the end of the P block
- “(iii) In Moens et al., the P block consensus seems to carry an exact duplication of the Q block at the end. This duplication has been kept in posterior works, such as in Burger-Calderón et al.” → this should be removed
- Furthermore, in Burger-Calderón et al. the sequence used for the R block is the same than that of the Q block+A, which is very surprising… I thus wonder whether it could be a copy/paste error from these authors.
L19 : “BKPyV subtyping has been mainly studied based on the genetic heterogeneity of the NCCR and VP1.” → to my knowledge, the NCCR is not used for BKPyV subtyping.
L110 : “giving rise” instead of “giving raise” ?
Other concerns :
I did not manage to run FASTA files. I can choose a file but when I click on “Analyse”, nothing happens.
Furthermore, I sometimes obtained this message but I don’t know why :
“Results from query 5f11c40e9f83b:
A PHP Error was encountered
Severity: Warning
Message: fopen(/var/www/bktyper.zidu.be/www/tmp/5f11c40e9f83b_results.csv): failed to open stream: No such file or directory
Filename: views/analyse.php
Line Number: 207”
Author Response
Dear authors, congratulations for the development of BKTyper. As previously mentioned, I think that this tool will be very useful for the BKPyV community. You will find below some comments on the V2 of your manuscript as well as your tool:
I still have some concerns concerning the section 3.2.1 and the Figure 2. As mentioned previously, Moens et al. defined the block length as O(142), P(68), Q(39), R(63) and S(63) (cf figure 1 of this reference), which is perfectly similar to the sequence used for BKTyper. Thus, there should be no misalignement between Moens and BKTyper in the Figure 2. Concerning the sequence mentioned by Burger-Calderon et al., I found that the block length was O(141), P(69), Q(38), R(39) and S(63). Thus:
- “(i) In Burger-Calderón et al. the first T of the O block is omitted (the first nucleotide of the origin of replication).” → OK
- “(ii) In Burger-Calderón et al., the first G of the Q block is missing.” → indeed it is placed at the end of the P block
- “(iii) In Moens et al., the P block consensus seems to carry an exact duplication of the Q block at the end. This duplication has been kept in posterior works, such as in Burger-Calderón et al.” → this should be removed
- Furthermore, in Burger-Calderón et al. the sequence used for the R block is the same than that of the Q block+A, which is very surprising… I thus wonder whether it could be a copy/paste error from these authors.
[Reply] Regarding the introductory point and point (iii), after a detailed look into Moens et al. 1995, we found the cause of discrepancy between the authors and the reviewer. In Moens et al., in Figure 1, blocks are summarized as O(142), P(68), Q(39), R(63) and S(63). However, in Figure 2, where sequences are displayed, P is of a 108 bp, having a layout of O(142), P(107), Q(39), R(63) and S(63). This P(107) it is made by P(68) + Q(39). This exact sequence was used during the development of BKTyper and of the manuscript, hence the disconcordance between the reviewer and this manuscript. In the previous submitted version, BKTyper in-text quotations of Moen’s P block were corrected, but not the alignments.
The sentence has been changed now from: “(iii) In Moens et al., the P block consensus seems to carry an exact duplication of the Q block at the end. This duplication has been kept in posterior works, such as in Burger-Calderón et al.” to “(iii) In Moens et al., there is a discrepancy on P length between Figure 1 [P(68)] and Figure 2 [P(107)]. In Figure 2 of the manuscript, the P block consensus seems to carry an exact duplication of the Q block at the end.”
Regarding point (iv), we appreciate the comment from the reviewer, and helping us to spot a second editing error, now in Burger-Calderon et al. We have decided to include it as point (iv) as “Finally, (iv), in Burger-Calderon et al. the R block is an exact copy of the Q block with an extra A at the end.”. Additionally, under Figure 2 caption, a clarification note has been added: “Editing errors from both Moens et al. and Burger-Calderón et al. are displayed in the alignment.”, to avoid further controversy.
L19 : “BKPyV subtyping has been mainly studied based on the genetic heterogeneity of the NCCR and VP1.” → to my knowledge, the NCCR is not used for BKPyV subtyping.
[Reply] The reviewer is technically right, VP1 is the BKPyV gene that is used for subtyping and so far, defines the genotypes. The NCCR is however a region of interest, and tends to be studied if not more, as frequently as VP1. We use the term typing/subtyping on a broader sense, as in “types of or kinds”, not as genotype deliberately. It is possible that our words may rise confusion, so the sentence has been changed to “BKPyV genetic diversity has been mainly studied based on the NCCR and VP1 […]” (line 19).
L110 : “giving rise” instead of “giving raise” ?
[Reply] We thank the reviewer to spot this mistake at line 81 (corrected).
Other concerns :
I did not manage to run FASTA files. I can choose a file but when I click on “Analyse”, nothing happens.
Furthermore, I sometimes obtained this message but I don’t know why :
“Results from query 5f11c40e9f83b:
A PHP Error was encountered
Severity: Warning
Message: fopen(/var/www/bktyper.zidu.be/www/tmp/5f11c40e9f83b_results.csv): failed to open stream: No such file or directory
Filename: views/analyse.php
Line Number: 207”
[Reply] We thank the reviewer for testing our tool. We have been catching up with the failed submissions of the first users (where we presume that both reviewers belong too), and we found that when there is a mistake in the fasta files (identation, deleted “>”, etc) the display the issue message is not displayed. It only displays the script error. We have fixed the webpage and once there is an input mistake, an appropriate informative message will be prompted so the users can go back and check their sequences.
Finally, we would like to acknowledge the constructive comments of the reviewers, in improving the quality of this work and helping to spot edit mistakes from Moens et al, Burger-Calderón et al and ourselves. We believe that reviewers work is seldomly acknowledged, albeit it is important for peer-review publications Therefore, we would like to include the reviewer in our acknowledgments section, for its constructive input on the subject.